# COTIC: Embracing Non-uniformity in Event Sequence Data via Multilayer Continuous Convolution

## Abstract

Massive samples of event sequences occur in various domains, including e-commerce, healthcare, and finance. There are two main challenges regarding modeling such data: methodological and computational. The methodological peculiarity for event sequences is their non-uniformity and sparsity. These requirements make time series models unsuitable. The computational challenge arises from a large amount of available data and the significant length of each sequence. Thus, the problem requires complex and efficient models. Existing solutions include large recurrent and transformer neural network architectures. On top of existing blocks, their authors introduce specific intensity functions defined at each moment. However, due to their parametric nature, these continuous-time-aware intensities represent only a limited class of event sequences.

We propose the COTIC method based on an efficient continuous convolution neural network suitable for the non-uniform occurrence of events in time. In COTIC, dilations and multi-layer architecture efficiently handle long-term dependencies between events. Furthermore, the model provides intensity dynamics in continuous time — including self-excitement encountered in practice. Being the first to introduce multiple continuous convolution layers that can handle arbitrary complex dependencies via MLP-modeled convolutions, we obtain these properties.

When benchmarked against existing models, the COTIC consistently outperforms them, especially in predicting the next event time and type: it has the average rank of 2.125 vs. 3.688 of the primal competitor. Additionally, its ability to produce effective embeddings showcases its potential for a range of downstream tasks, as produced embeddings are sufficient to solve various downstream tasks, e.g., 0.459 vs. 0.452 baseline accuracy on a 4-label age bin prediction for transactions dataset. The code of the proposed method is available at https://anonymous.4open.science/r/COTIC-F47D/README.md

Table 1: Ranks for methods averaged over eight datasets for the problem of next event time and type predictions. A lower rank means that the method is closer to the top-1 method (with the rank one) for a problem. The best results are highlighted with bold font, and the second-best results are underlined.

| Method | Next event time MAE, rank | Next event type Accuracy, rank | Mean rank |
|---|---|---|---|
| RMTPP Du et al. (2016) | 3.625 | 6.500 | 5.063 |
| Neural Hawkes Mei & Eisner (2017b) | 4.125 | 3.250 | 3.688 |
| ODETPP Chen et al. (2021) | 7.000 | 6.000 | 6.500 |
| THP Zuo et al. (2020) | 7.125 | 5.250 | 6.188 |
| THP2SAHP Zhang et al. (2020) | 6.750 | 5.000 | 5.875 |
| Attentive NHP Yang et al. (2022) | 3.625 | 4.750 | 4.188 |
| WaveNet van den Oord et al. (2016) | 4.875 | 3.000 | 3.938 |
| CCNN Shi et al. (2021) | 5.750 | 8.625 | 7.188 |
| COTIC (ours) | **2.125** | **2.625** | **2.375** |

# 1 INTRODUCTION

A lot of real-world processes are event sequences: bank transactions Wei et al. (2013), purchases at a store Lysenko et al. (2019), network traffic Saha et al. (2019), series of messages, social media posts, views of TV programs, flows of customers, etc. Consequently, there is a growing need to model such data, e.g., one may be interested in predicting the banking activity of a particular customer or estimating the timing of the next purchase made at an online marketplace. This knowledge provides new perspectives for a business: for instance, a churn prediction Berger & Kompan (2019). Timely churn prediction straightforwardly leads to more precise and effective marketing campaigns.

For all domains and problems, we can say that the next event occurs at a random time after a previous event. So, a whole sequence is non-uniform and discrete, as we have event occurrence information only at a finite number of time points. Moreover, the label of an event is another random variable that can depend on the past. These non-uniformity and discreteness require specific formalism to make modeling possible. Specifically, event sequences are realizations of temporal point processes (TPP). The key concept for TPPs is the intensity, the expectation of the number of events of each type observed during a small period of time. For textbook examples of TPP, the homogeneous and non-homogeneous Poisson processes Lawless (1987); Kingman (1992), the intensity of the process is independent of past events. It is not the case for the spread of a pandemic or a history of orders. A more realistic model is a self-exciting TPP or the Hawkes process Hawkes (1971). For them, previous events influence the future intensity of the event flow.

The diversity of available data makes the event sequence modeling complex and requires models to be general, as different cases have their peculiarities. Constant advances in deep learning make researchers look for neural network architectures to model event sequences. LSTM-like architectures were the first to be proposed in Mei & Eisner (2017b); Du et al. (2016). Different approach, that combined Neural Jump ODEs with Continuous Normalizing Flows, was proposed in Chen et al. (2021). These methods were accompanied by later Transformer-based architecture in Zuo et al. (2020); Yang et al. (2022). The study Shi et al. (2021) utilized convolutional neural networks (CNNs) for non-uniform time series, while showing sub-optimal performance to model event sequences due to usage of a single continuous convolution layer and manual grid allocation for further convolutional layers. While making decent progress, all these models have an important limitation: they recover the intensity function within a fixed exponential or semi-linear parametric family and utilize it to derive the following event type or the next event time prediction.

In this work, we advance the idea of exploring a class of modeled event sequences by proposing a COntinuous-TIme Convolutional neural network model (COTIC). It constructs representations of a temporal point process, predicts its intensity, and solves other downstream problems. Our contributions are the following:

- Our novel COTIC, a deep one-dimensional continuous CNN, processes event sequences, that has non-uniform in time structure. Due to modeling of a convolution with a MLP of time and similar innovation of the output layer, the model abstains from the usage of a closed-form parametric assumption for intensity. Thus, we are able to model a large variety of possible dependencies between events. See Figure 2 for the constructed intensity function example.

- In benchmark tests on eight datasets, our method outperforms existing approaches — including Transformers, RNNs, Neural ODEs, and baseline CNNs — in classic problems of predicting time and type of next event. Table 1 demonstrates the superiority of COTIC for both problems. Moreover, COTIC's training time is better than that of the nearest pursuers with comparable performance.

- Thanks to COTIC's generative nature, it offers self-supervised representation learning capabilities. We compared the embedding quality on other event sequence methods as well as a standard time series classification approach MiniRocket Dempster et al. (2021) on the transactions dataset for the age bin prediction. The accuracy for a logistic regression model based on COTIC's embeddings is superior to alternatives.

## 2 RELATED WORK

We refer an interested reader for the overview of the current state of the neural network models for modeling TPPs to the review Shchur et al. (2021). Here, we provide more details of the most relevant papers to put our work in context.

**Temporal point processes (TPPs).** For a *temporal* point process, each point lies on a temporal $\mathbb{R}^1$ axis. Consequently, there is a timestamp associated with each point. For a *marked* temporal point process, we have an additional mark or a vector of features associated with each point. We can say that the vector of features belongs to $\mathbb{R}^d$ Yan (2019). So, event sequences are a particular case of realizations of marked temporal point processes. Self-exciting point processes are of specific interest, as they assume that the future intensity of events depends on history. Typically, past events increase intensity in the future — for example, it is the case for retweets in social networks Rizoiu et al. (2017) or the spread of a virus Chiang et al. (2022).

**Deep learning for TPPs.** While classic machine learning succeeds in modeling complex dynamics of event sequences Zhou et al. (2013), the introduction of deep learning leads to more adaptability and, consequently, better results Mei & Eisner (2017a). On the contrary, the main downside of deep learning is higher computational costs to train such models Zhuzhel et al. (2021).

A variety of deep learning architectures results in a multitude of ways to model temporal point processes. Historically, the first papers considered different recurrent architectures like Continuous LSTM Mei & Eisner (2017a) or RMTPP Du et al. (2016). Adaptation of transformer architectures followed via e.g. Zuo et al. (2020); Zhang et al. (2020). The convolutional neural network has been considered indirectly in Shi et al. (2021). Their architecture had only two layers, with only the first one being responsible for time transformation from non-uniform to uniform, so they didn't receive SOTA results for TPP modeling. The papers on state-space layers Gu et al. (2022) also mention the non-uniformity of time as one of the challenges while not providing a recipe for an efficient non-uniform time state-space layer.

We also note that TPP-based learning happens in a self-supervised way without additional label information, as we can get event types and event occurrence times from the data itself. However, direct approaches for treating sequential data in a self-supervised way for time series and event sequences based on contrastive Yue et al. (2022); Babaev et al. (2022), non-contrastive Marusov & Zaytsev (2023) and generative strategies Kenton & Toutanova (2019) are complementary to TPPs.

**Deep learning for sequential data.** Within the community engaged in sequential data modeling, a lively discussion persists regarding the most effective architecture among Transformers, Recurrent Neural Networks, and Convolutional Neural Networks.

Transformers often dominate the discourse surrounding sequential data, given their remarkable performance in Natural Language Processing. Nonetheless, it remains an open question whether Transformers are optimal for, e.g., time-series modeling. Supporting their use, certain transformer-based architectures have been designed to incorporate time-series specific characteristics, such as trend and seasonality, as demonstrated in Wu et al. (2021). However, Transformer remains a relatively inefficient approach with a squared complexity on the sequence length. Other counterarguments exist, with some researchers contending that, despite the use of positional encoding, Transformers' inherent permutational-invariance could lead to a loss of crucial information, as per Zeng et al. (2023).

Furthermore, certain studies have suggested the superiority of Convolutional Neural Networks (CNNs) in addressing specific problems. For instance, the issue of a constrained CNN horizon can be circumvented by implementing continuous convolutions, thereby enabling the processing of non-uniform data as proposed by Romero et al. (2022).

**Research gap.** Careful examination of the domain of event sequence data can help with identification of the best approach with respect to quality and efficiency. The main challenge for applying convolutional neural networks as models for TPPs is the non-uniformity and discreteness of corresponding sequences. Instead of learning values at discrete points of a convolutional kernel, we should learn the kernel as a whole. Recent results demonstrate that this scenario is admissible Romero et al. (2021) for multivariate time series and non-uniform observations for time series Shi et al. (2021). However, due to these models' constraints discussed above they don't suit for

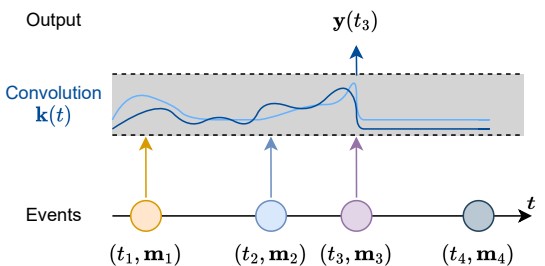

Output      $\mathbf{y}(t_3)$

Convolution $\mathbf{k}(t)$

Events

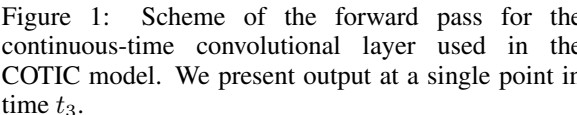

$(t_1, \mathbf{m}_1)$    $(t_2, \mathbf{m}_2)$   $(t_3, \mathbf{m}_3)$     $(t_4, \mathbf{m}_4)$

Figure 1: Scheme of the forward pass for the continuous-time convolutional layer used in the COTIC model. We present output at a single point in time $t_3$.

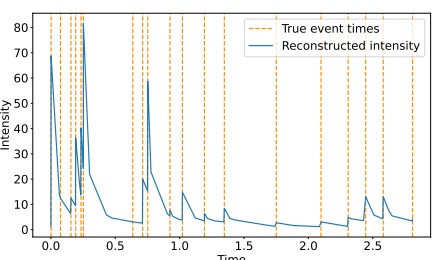

Figure 2: The intensity function reconstructed by the COTIC model for a sequence from the MIMIC dataset. We present the sum of intensities corresponding to all of the 85 event types.

high quality TPP modeling. We aim to produce a model based on a convolutional architecture that would be accurate and fast both in training and inference, handling long-term dependencies, and solving various downstream tasks via produced embeddings.

## 3 METHODS

In this section, we describe our Continuous-time Convolutions (COTIC) model and highlight its differences from other neural models for the processing of event sequence data.

### 3.1 PRELIMINARIES

**Notation** Suppose we observe $n$ sequences in a sample $D = \{\boldsymbol{S}_i\}_{i=1}^n$. Each sequence consists of tuples $\boldsymbol{S}_i = \{(t_{ij}, \mathbf{m}_{ij})\}_{j=1}^{T_i}$, where $t_{ij} \in [0, T_i]$ are event times, such that for $k > j$ $t_{ik} \geq t_{jk}$, and $\mathbf{m}_{ij} \in \mathbb{R}^d$ denote event type marks for the $i$-th sequence in a dataset. If we have a discrete set of possible marks $m_{ij} \in \{1, \ldots, M\}$, we further work with an embedding of each mark instead. We use the notation $\boldsymbol{S} = \{(t_j, \mathbf{m}_j)\}_{j=1}^T$ when an arbitrary sequence from the dataset is meant. For the current event number $k$, we denote the observation's history as $\boldsymbol{S}_{1:k} = \{(t_j, \mathbf{m}_j)\}_{j=1}^k$ for $k$ first events and $\boldsymbol{S}_{<t}$ for all events with $t_i < t$.

**Intensity function and likelihood** When dealing with event sequences, there is a question: how to process the non-uniform nature of time lags between events? One of the approaches is to use the intensity function.

One can consider return time $\tau_i = t_i - t_{i-1}$ and event type $m$ as random variables. Then, one can introduce its conditional probability density function (PDF) $f(t, m)$, which shows the probability of the event of type $m$ happening in an infinitely small time interval, and the conditional Survival Function $S(t)$, equal to the probability that the event of arbitrary type has not happened till the moment of time $t$:

$$f(t = t_i, m = m_i | \boldsymbol{S}_{1:i-1}) = \lim_{\Delta t \to +0} \frac{\mathbb{P}(t < t_i \leq t + \Delta t, \, m = m_i | \boldsymbol{S}_{1:i-1})}{\Delta t}, \tag{1}$$

$$S(t) = \mathbb{P}(\tau_i \geq t | \boldsymbol{S}_{1:i-1}) = \int_t^\infty f(z)dz. \tag{2}$$

We denote the conditional intensity function of the sequence:

$$\lambda_m(t) = \mathbb{E}\left[\frac{dN_m(t)}{dt}\right] = \frac{f(t, m | \boldsymbol{S}_{<t})}{S(t)}, \tag{3}$$

where $N_m(t) = \#\{t_j : t_j < t | m_j = m\}$ is the number of events of type $m$ occurred before time $t$. The main benefit of using the intensity function instead of the PDF is that it has fewer restrictions.

The intensity has to be non-negative and should lay in the space of $L_1$ functions. There is no need to check that it integrates to 1 as it would be if we were to work with a PDF.

At the same time, it is easy to express the likelihood in terms of the intensity function. For a vector parametric intensity function $\boldsymbol{\lambda}(t|\boldsymbol{\theta}) = \{\lambda_m(t|\boldsymbol{\theta})\}_{m=1}^M$ with the vector of parameters $\boldsymbol{\theta}$, the negative log-likelihood for an event sequence $S_{1:k}$ with no event in the interval $(t_k, T]$ is:

$$L_\lambda(\boldsymbol{\theta}) = \int_0^T \sum_m \lambda_m(t|\boldsymbol{\theta})dt - \sum_{j=1}^k \log \lambda_{m_j}(t_j|\boldsymbol{\theta}). \tag{4}$$

Once the intensity $\boldsymbol{\lambda}(t|\boldsymbol{\theta})$ is defined, it is possible to optimize this model via likelihood maximization. As the considered one-dimensional integral doesn't have an analytical form, its Monte-Carlo estimator is used instead.

## 3.2 PROBLEM STATEMENTS

For an event sequence, we consider intensity function reconstruction as the main problem. To check the quality of it, we consider two tasks on top of it: return time and event type prediction.

**Return time prediction** The goal is to predict the time difference $\Delta t_{k+1} = t_{k+1} - t_k$ from the current event $k$ to the next one $t_{k+1}$ using $\boldsymbol{S}_{1:k}$.

**Event type prediction** In our set-up, events belong to $M$ distinct types. Thus, we aim to predict the type of the next event $\mathbf{m}_{k+1}$ using $\boldsymbol{S}_{1:k}$.

## 3.3 CONTINUOUS CONVOLUTION OF EVENT SEQUENCES

As the time intervals $[t_j, t_{j+1}]$ are not necessarily equal to each other, the considered time series are non-uniform. Standard 1-dimensional CNNs are designed for equally lagged data. However, we can extend this idea to a non-uniform case. The considered event sequence $\boldsymbol{S}$ can be rewritten in the functional form:

$$\mathbf{m}(t) = \sum_{j=1}^k \mathbf{m}_j \delta(t - t_j), \tag{5}$$

where $\delta(\cdot)$ is the Dirac delta function.

Let $\boldsymbol{\kappa}(t)\colon \mathbb{R} \to \mathbb{R}^{d_{out} \times d}$ be a kernel function. In order to make the model causal and prevent peeking into the future, let's state $\boldsymbol{\kappa}(t) = 0$, $t < 0$. The continuous convolution is:

$$\mathbf{y}(t) = (\boldsymbol{\kappa} * \mathbf{m})(t) = \int_0^\infty \boldsymbol{\kappa}(t - \tau)\mathbf{m}(\tau)d\tau = \sum_{j=1}^k \boldsymbol{\kappa}(t - t_j)\mathbf{m}_j. \tag{6}$$

The last equality holds due to the specific form of $\mathbf{m}(t)$ given in equation 5.

In our neural network layer, we can use the vector function of time $\mathbf{y}(t)$ in two ways. The first way corresponds to our need to get $\mathbf{y}(t)$ at some specific point. In this case, we query $\mathbf{y}(t)$ at this moment of time to have a representation. The second way corresponds to the usage of $\mathbf{y}(t)$ as an input to the next convolutional layer. In this case, we limit $\mathbf{y}(t)$ to the set $\{t_1, \ldots, t_k\}$ obtaining a sequence of vectors $\{\mathbf{y}_1, \ldots, \mathbf{y}_k\}$ with $\mathbf{y}_i = \mathbf{y}(t_i)$. Now we use the functional form equation 5 to have an input to the next layer. The scheme of this continuous convolutional layer is presented in Figure 1. This approach allows us not to lose the temporal structure for all the layers of the model and query the embedding at any point in time.

## 3.4 COTIC MODEL ARCHITECTURE

Continuous one-dimensional convolution introduced by equation 6 allows us to get a new continuous representation of an input event sequence. We parameterize the convolution kernel $\boldsymbol{\kappa}$ by a fully-connected neural network and stack several such convolutions to obtain embeddings of a considered sequence.

The model receives embeddings at the final layer. It applies $L$ convolution layers to the input $\mathbf{m}(t)$ with non-linearities $\sigma$ being the LeakyReLU activation:

$$\mathbf{h}_{i,1:k} = \sigma\left(\boldsymbol{\kappa}_{\theta_L} * \sigma\left(\ldots * \sigma\left(\boldsymbol{\kappa}_{\theta_2} * \sigma\left(\boldsymbol{\kappa}_{\theta_1} * \mathbf{m}(t)\right)\right)\ldots\right)\right).$$

To reduce the computational complexity on each layer, we use a limited kernel size that restricts the number of previous points taken into account. Also, we use dilated convolution to increase the receptive field. So, the result of the application of $L$ convolutions layers to the sequence of events $S_{i,1:k}$ is the matrix $H_{i,1:k}$ that consists of embeddings for each event $\mathbf{h}_{ij} \in \mathbb{R}^d$. The embeddings induce the function $\mathbf{h}(t)$ from equation 5.

Now we are ready to produce the intensity function $\hat{\boldsymbol{\lambda}}(t)$, expected event type $\widehat{m}_{i,k+1}$, and expected return time $\Delta\widehat{t}_{i,k+1}$ correspondingly via the following formulas:

$$\hat{\boldsymbol{\lambda}}(t) = \mathrm{softplus}\left(\mathrm{Linear}\left(\sigma\left(\boldsymbol{\kappa}_{\theta_\lambda} * \mathbf{h}(t)\right)\right)\right), \tag{7}$$

$$\widehat{m}_{i,k+1} = \mathrm{MLP}_2\left(\mathbf{h}_{i,k}\right), \Delta\widehat{t}_{i,k+1} = \mathrm{MLP}_1\left(\mathbf{h}_{i,k}\right), \tag{8}$$

here $\mathrm{MLP}_i$ are multi-layer perceptron prediction head networks, the estimated vector intensity $\hat{\boldsymbol{\lambda}}(t)$ has the output dimensional equal to the number of possible event types.

To sum up, once the embeddings $\mathbf{h}_i$ of a sequence are obtained by a $L$-layer Continuous CNN, they are fed into MLPs to produce the final predictions. We use one additional continuous convolution followed by a linear layer to estimate the intensity function of the process at an arbitrary time step.

### 3.5 Loss functions and training pipeline

The proposed model is trained with a combined loss function consisting of several terms. In order to train the convolutional part of the model, we use *the negative log-likelihood $L_\lambda(\boldsymbol{\theta})$* equation 4 as the loss function. The prediction heads equation 8 are trained with the standard LogCosh and CrossEntropy losses, respectively:

$$L_{time}\left(\Delta\hat{t}_{k+1}, \Delta t_{k+1}\right) = \mathrm{LogCosh}\left(\Delta\hat{t}_{k+1} - \Delta t_{k+1}\right) =$$

$$= \left(\Delta\hat{t}_{k+1} - \Delta t_{k+1}\right) + \log\left(1 + e^{-2\left(\Delta\hat{t}_{k+1} - \Delta t_{k+1}\right)}\right),$$

$$L_{type}\left(\widehat{m}_{k+1}, m_{k+1}\right) = \mathrm{CrossEntropy}\left(\widehat{m}_{k+1}, m_{k+1}\right) = -\sum_{l=1}^{K} m_{k+1}^{(l)} \log\frac{\exp\left(\widehat{m}_{k+1}^{(l)}\right)}{\sum_{c=1}^{K}\exp\left(\widehat{m}_{k+1}^{(c)}\right)},$$

where $m_{k+1}^{(l)}$ are the labels for the true event type and $\widehat{m}_{k+1}^{(l)}$ are the predicted event type scores for the $(k+1)$-th event in a sequence. Note that the loss functions above are calculated for a single sequence. Given a batch of sequences, the losses are averaged. The resulting loss function for the prediction head training is a weighted sum: $L_{heads} = \alpha L_{time} + \beta L_{type}$.

The training pipeline for the model is the following. We first train its convolutional part with the log-likelihood loss for $N_0$ epochs while the prediction heads are frozen. Afterward, we continue training the whole model jointly with the combined loss $L = L_\lambda + L_{heads}$, the gradient for $L_{heads}$ doesn't go the main body of the model.

## 4 Experiments

### 4.1 Datasets

We employ a variety of event sequence datasets to benchmark our method against the others. Comprehensive statistics for these eight datasets are demonstrated in Table 2. In Appendix's section A, we delve deeper to obtain additional insights about each of them.

### 4.2 Details of comparison

For all the experiments, we split the datasets into three parts (train set, test set, and validation set) in the ratio $8:1:1$. We train the models via Adam Kingma & Ba (2015) optimizer for the maximum

Table 2: Statistics of the used datasets

| Dataset | # of event types | Mean sequence length | # of sequences |
|---|---|---|---|
| Retweet | 3 | 108.8 | 29000 |
| Amazon | 8 | 56.5 | 7523 |
| IPTV | 16 | 3225.2 | 302 |
| SO | 22 | 72.4 | 33165 |
| Transactions | 60 | 862.4 | 30000 |
| Mimic | 65 | 8.6 | 285 |
| LinkedIn | 82 | 3.1 | 2439 |
| MemeTrack | 4977 | 10.8 | 29696 |

Table 3: Return time prediction MAE ↓ results. The best result for each dataset is in bold; the second-best model result is underlined.

| Dataset | RMTPP | Neural Hawkes | ODETPP | THP | THP2SAHP | AttNHP | WaveNet | CCNN | COTIC |
|---|---|---|---|---|---|---|---|---|---|
| Retweet | $\underline{0.030 \pm 0.000}$ | $\mathbf{0.029 \pm 0.001}$ | $144.707 \pm 140.205$ | $0.043 \pm 0.014$ | $0.060 \pm 0.019$ | $\mathbf{0.029 \pm 0.000}$ | $0.224 \pm 0.021$ | $0.138 \pm 0.027$ | $0.034 \pm 0.002$ |
| Amazon | – | $89.320 \pm 3.320$ | $62381 \pm 3338$ | $113.470 \pm 4.180$ | $98.750 \pm 8.400$ | $347.344 \pm 0.047$ | $\underline{41.950 \pm 0.600}$ | $58.900 \pm 0.010$ | $\mathbf{37.810 \pm 2.510}$ |
| IPTV | $\mathbf{0.098 \pm 0.001}$ | $\underline{0.099 \pm 0.001}$ | $78003 \pm 24645$ | $0.402 \pm 0.029$ | $0.687 \pm 0.077$ | $\mathbf{0.098 \pm 0.001}$ | $1.280 \pm 0.680$ | $0.111 \pm 0.007$ | $0.123 \pm 0.001$ |
| SO | $0.839 \pm 0.001$ | $0.841 \pm 0.001$ | $0.843 \pm 0.001$ | $13.480 \pm 0.840$ | $11.980 \pm 0.270$ | $0.857 \pm 0.020$ | $\underline{0.676 \pm 0.008}$ | $1.000 \pm 0.070$ | $\mathbf{0.641 \pm 0.001}$ |
| Transactions | $0.842 \pm 0.001$ | $2.072 \pm 0.709$ | $35.147 \pm 59.416$ | $1.480 \pm 0.350$ | $1.240 \pm 0.060$ | $0.839 \pm 0.001$ | $0.688 \pm 0.010$ | $\underline{0.821 \pm 0.004}$ | $\mathbf{0.595 \pm 0.006}$ |
| Mimic | $\underline{0.354 \pm 0.001}$ | $0.415 \pm 0.041$ | $0.367 \pm 0.007$ | $0.584 \pm 0.013$ | $0.518 \pm 0.026$ | $0.365 \pm 0.018$ | $3.280 \pm 1.140$ | $0.555 \pm 0.022$ | $\mathbf{0.339 \pm 0.004}$ |
| LinkedIn | $2.521 \pm 0.001$ | $2.483 \pm 0.015$ | $6.057 \pm 4.371$ | $9.890 \pm 0.290$ | $9.520 \pm 0.220$ | $2.469 \pm 0.002$ | $\underline{1.490 \pm 0.020}$ | $2.560 \pm 0.350$ | $\mathbf{1.380 \pm 0.010}$ |
| MemeTrack | – | $46.803 \pm 0.009$ | $59.759 \pm 12.409$ | $97.770 \pm 0.650$ | $99.690 \pm 8.690$ | $\mathbf{46.359 \pm 0.154}$ | $63.080 \pm 1.460$ | $138.470 \pm 9.180$ | $\underline{46.710 \pm 0.020}$ |
| # of wins | 1 | 1 | 0 | 0 | 0 | $\underline{3}$ | 0 | 0 | $\mathbf{5}$ |

Table 4: Event type prediction accuracy ↑ results. The best result for each dataset is in bold; the second-best model result is underlined.

| Dataset | RMTPP | Neural Hawkes | ODETPP | THP | THP2SAHP | AttNHP | WaveNet | CCNN | COTIC |
|---|---|---|---|---|---|---|---|---|---|
| Retweet | $0.555 \pm 0.010$ | $0.605 \pm 0.003$ | $0.525 \pm 0.025$ | $0.499 \pm 0.031$ | $0.518 \pm 0.022$ | $0.578 \pm 0.004$ | $\underline{0.606 \pm 0.001}$ | $0.349 \pm 0.043$ | $\mathbf{0.608 \pm 0.002}$ |
| Amazon | – | $\mathbf{0.872 \pm 0.009}$ | $0.041 \pm 0.008$ | $0.331 \pm 0.005$ | $0.331 \pm 0.006$ | $0.189 \pm 0.082$ | $\underline{0.564 \pm 0.005}$ | $0.095 \pm 0.021$ | $0.550 \pm 0.050$ |
| IPTV | $0.224 \pm 0.016$ | $0.774 \pm 0.006$ | $0.198 \pm 0.046$ | $\underline{0.779 \pm 0.001}$ | $\mathbf{0.784 \pm 0.000}$ | $0.430 \pm 0.009$ | $0.776 \pm 0.001$ | $0.061 \pm 0.022$ | $0.734 \pm 0.026$ |
| SO | $0.366 \pm 0.117$ | $0.431 \pm 0.017$ | $0.444 \pm 0.005$ | $0.432 \pm 0.001$ | $0.432 \pm 0.001$ | $0.327 \pm 0.063$ | $\mathbf{0.470 \pm 0.000}$ | $0.056 \pm 0.010$ | $\underline{0.463 \pm 0.001}$ |
| Transactions | $0.252 \pm 0.109$ | $0.293 \pm 0.026$ | $0.305 \pm 0.018$ | $0.315 \pm 0.000$ | $0.315 \pm 0.000$ | $0.189 \pm 0.082$ | $\underline{0.369 \pm 0.011}$ | $0.017 \pm 0.003$ | $\mathbf{0.346 \pm 0.015}$ |
| Mimic | $0.874 \pm 0.034$ | $\mathbf{0.930 \pm 0.012}$ | $0.855 \pm 0.029$ | $0.699 \pm 0.176$ | $0.788 \pm 0.067$ | $0.788 \pm 0.067$ | $0.300 \pm 0.250$ | $0.013 \pm 0.008$ | $\underline{0.862 \pm 0.029}$ |
| LinkedIn | $0.148 \pm 0.002$ | $0.255 \pm 0.004$ | $0.210 \pm 0.010$ | $0.132 \pm 0.009$ | $0.147 \pm 0.008$ | $\mathbf{0.278 \pm 0.008}$ | $0.262 \pm 0.011$ | $0.007 \pm 0.001$ | $\underline{0.270 \pm 0.020}$ |
| MemeTrack | – | $\mathbf{0.117 \pm 0.037}$ | $0.010 \pm 0.008$ | $0.013 \pm 0.002$ | $0.014 \pm 0.001$ | $0.051 \pm 0.009$ | $0.050 \pm 0.004$ | $0.005 \pm 0.001$ | $\underline{0.072 \pm 0.008}$ |
| # of wins | 0 | $\mathbf{3}$ | 0 | 0 | 1 | 1 | $\underline{2}$ | 0 | 1 |

100 epochs. Among existing approaches, to compare it with COTIC, we consider top-performers from various architectures using TPP-based RNNs (RMTPP, Neural Hawkes, ODETPP), TPP-based transformers (THP, THP2SAHP, AttNHP) and adoption of existing CNN to the processing of non-uniform event sequences (WaveNet, CCNN). The details of these methods are given in Appendix C accompanied by technical details on used implementations in Appendix C.1.

### 4.3 MAIN EXPERIMENTS

**Return time and event type prediction.** We have compared the COTIC model with the baselines on two problems commonly used to evaluate the quality of event sequences models: return time and event type prediction. It is worth noting that likelihood computation could vary between methods, and these values cannot be used to compare different models. For all the experiments, we average results and estimate standard deviations using five independent runs.

*Return time prediction.* Table 3 presents obtained results. Our method is either the best or close to the second-best model regarding the return time estimation. Moreover, it is winning five times out of eight, and there is no obvious competitor because three models tie for second place, thus not performing well consistently.

*Event type prediction.* Our findings regarding the event type prediction experiment are presented in Table 4. The COTIC model exhibits high performance, being competitive against the other contenders. The critical aspect to note is the superior mean rank of our model, presented in Table 1. This reflects its enhanced predictive abilities and consistency across various presented samples. While the Neural Hawkes model recorded the highest accuracy in 3 datasets out of 8, its high computational cost can render it less suitable for event sequences processing tasks.

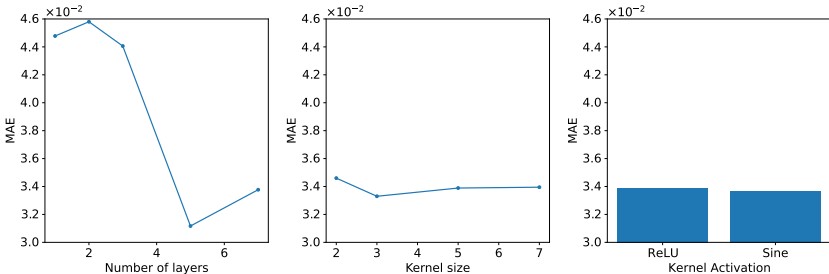

Figure 3: Sensitivity study results. We vary the number of layers, the kernel size, and the activation function (from left to right) to obtain MAE values for the return time prediction on the Retweet dataset.

Additional experiments in Appendix D suggest that these conclusions hold for various available event types and sequence lengths. These experiments also demonstrate our model's and transformers' stability, while the RNN-based method shows less stable results.

**Downstream problem** Neural TPPs are generative models, thus, we expect them to have decent representation capabilities. Consequently, we can reuse the obtained representation for other downstream problems. To check this, we conduct an additional experiment. For the Transactions dataset, we predict one of four age bins using embeddings obtained from a model. For each model, we train Logistic Regression on top of the embeddings, performing linear probing with a frozen model Li et al. (2021). We compared COTIC with $64$ embedding size to other intensity-based models and the MiniRocket model with $256$ kernels Dempster et al. (2021) designed for uniform time series classification. For the MiniRocket, time as a feature and one-hot encoded event types are inputs for this model.

The results are in Table 5. Our model showed the best results among TPP-based embeddings, and MiniRocket is slightly inferior to the COTIC. We hypothesize that other TPP-based approaches provide very local representation of an event sequence at each points, while our convolutional layers lead to larger representative power.

**Discussion.** On average, regarding the obtained ranks in Table 1, the COTIC model shows the best performance followed by Neural Hawkes, WaveNet and Attentive Neural Hawkes. Return time results indicate that convolutions are very powerful in processing event sequences, while recurrent models, like Neural Hawkes, are useful for event-type prediction. In addition, training time of Neural Hawkes and Attentive Neural Hawkes is prohibitive for many datasets. CCNN Shi et al. (2021), the closest solution to ours, underperforms both for the event type and return time prediction problems, because it was designed for different problems. Thus, a careful introduction of multiple non-uniform convolutional layers is essential to model event sequences via CNN.

## 4.4 SENSITIVITY STUDY

Our previous claims highlighted that a high number of convolutional layers for COTIC is vital for high performance. We also want to verify what kernel size and activation function we need. These experiments were run on the Retweet dataset; the results are shown in Figure 3.

There is a tangible trade-off on the number of layers: if the number of layers is too low, then the model's complexity and the receptive field size are not enough to catch data intricacies; if we make the model too deep, it becomes harder to train it, ending up with a slightly underperforming model. However, there is little dependence on the kernel size and the activation function. The first result means that, given enough layers, the model already has a large enough receptive field and complexity

Table 5: Age bin prediction accuracy for Transactions dataset

| Method | Base | Neural Hawkes | THP | ODETPP | ATTNHP | MiniRocket | COTIC (ours) |
|---|---|---|---|---|---|---|---|
| Accuracy | 0.25 | 0.254 | 0.290 | 0.265 | 0.258 | 0.452 | **0.459** |

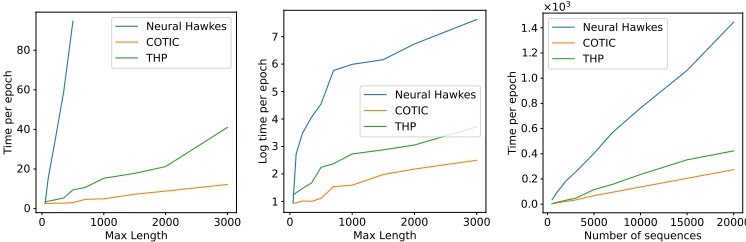

Figure 4: Dependence of training time complexity for different approaches on the sequence length (on IPTV dataset) and the number of sequences (on SO dataset). Sequence length results are presented in two plots, the first one is for time, and the second is for the time logarithm.

to catch all the temporal dependencies within the dataset, and there is no need to make the model even more complex by increasing the kernel size. Despite that, one should be careful in interpolating this fact to all the other datasets as soon it can be the property of particular retweet sequences. The latter experiment was to substitute the ReLU in the kernel with a new Sine activation Sitzmann et al. (2020). As one can notice, there was almost no influence on the performance.

## 4.5 TIME COMPLEXITY

Analyzing time complexity, i.e., the dependence of the algorithm training time concerning some dataset properties, is essential to understanding the scalability of the methods. Thus, we conducted additional experiments and measured training time per epoch based on the maximal sequence length and the number of sequences in a dataset. We used NVIDIA GeForce GTX 1080 Ti graphics cards to run all the experiments in this section. For some additional experiments we used NVIDIA GeForce A10 graphics cards to run all the experiments.

The outcomes of this experiment are depicted in Figure 4. Theoretically, the time complexity is anticipated to be linear with respect to the number of sequences, which is, indeed, proved in practice. However, the dynamic relating time complexity to sequence length is more intricate. This complexity arises due to our method of truncating sequences only when they surpass a specific length, as the distribution of sequence lengths is non-uniform.

Additional results for other models on their time complexity and the number of parameters are given in Appendix E. They provide additional insight into our claim that COTIC has superior training time compared to models with similar performance.

## 5 CONCLUSION

Modeling temporal point processes is a long-standing challenge, and our research sought to address this using a novel approach. We leverage a CNN to model temporal point processes, resulting in the development of our COntinuous-TIme Convolutional (COTIC) model of event sequences.

By introducing a continuous convolution function for non-uniform sequences, we propose an approach that holds several inherent advantages. These include the utilization of dilation for long-term memory, separate convolutions for various event types, and a variable depth allowing for smooth time transformations, which is required to process non-uniform event sequences.

The experimental results highlight the quality and efficiency of the COTIC model. It outperforms both recurrent and transformer-based methods on datasets routinely employed to benchmark temporal point process models. Additionally, the COTIC offers greater efficiency concerning training time in comparison to methods with similar performance. Given its robust performance and efficiency, the COTIC model has the potential to set a new benchmark in temporal point process modeling, with numerous possible applications that utilize event sequences.

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

## A DETAILS ON DATASETS

**Retweets** Zhao et al. (2015): The dataset presents sequences of tweets containing an origin tweet and some follow-up tweets. We record the time and the user tag of each tweet. Further, users are grouped into three categories based on the number of their followers: "small", "medium", and "large". These groups correspond to the three types of events.

**Amazon** ama (2018): This dataset consists of around 7.5k user sequences with product reviews and product categories. All products are divided into 8 distinct groups.

**IPTV** Luo et al. (2014): The dataset contains 7.1k IPTV users viewing records from Shanghai Telecomm Inc, categorizing TV programs into 16 types. In addition, the dataset includes viewing patterns for sessions longer than 20 minutes.

**StackOverflow** Leskovec & Krevl (2014): StackOverflow (SO) is a question-answering website. It rewards users with badges to promote engagement in the community, and the same badge can be given multiple times to the same user. We collect data for a two-year period and treat each user's reward history as a sequence. Each event in the sequence signifies a receipt of a particular medal.

**Transactions** Fursov et al. (2021): The dataset contains sequences of transaction records stemming from clients of financial institutions. For each client, we have a sequence of transactions. We describe each transaction with a discrete code, identifying the type of transaction, and the Merchant Category Code, such as a "bookstore", "ATM", "drugstore", etc.

**MIMIC-II (Electrical Medical Records)** Johnson et al. (2016): MIMIC collects patients' visits to a hospital's ICU in a seven-year period. We treat the visits of each patient as a separate sequence where each element in the sequence contains a time stamp and a diagnosis.

**LinkedIn** Xu & Zha (2017): This dataset provides information on users' sequences of employment companies. There are 82 different companies comprising different event types with 2439 users in total.

**MemeTrack** Leskovec & Krevl (2014): It collects mentions of 42k different memes spanning ten months, combining data from over 5000 websites with over 1.5 million documents including blog posts, web articles, etc. Each sequence in this dataset is the life-cycle of a particular meme where each event (meme occurrence) is associated with a timestamp and a website ID, serving as the event type.

## B  TECHNICAL DETAILS FOR COTIC

In this part, we provide some details of COTIC's training.

The description of the full architecture is given in section 3. The kernel functions at each layer $\kappa_{\theta_l}(t)$ were modeled by three-hidden-layer MLPs with hidden sizes equal to $8$, $4$, and $8$, respectively.

Other used hyperparameters can be found in Table 6. For all datasets, we started with a default set of hyperparameters and slightly tuned them. Note, that we consider two alternative variants of the number of layers: for some datasets a better option is a deep 9-layers convolutional architecture, for others 2 layers are sufficient. We have used the Adam optimizer with the initial learning rate of $0.1$ and the multi-step scheduler with $\gamma = 0.1$ and steps on the 40-th and 75-th epoch.

The MAE for the return time prediction for different configurations of hyperparameters for LinkedIn and MIMIC datasets are given in Table Figure 5 for configurations obtained during hyperparameter search via Optuna Akiba et al. (2019). The performance is pretty strong for different variants, while we can get additional improvement with hyperparameteres tuning.

| Hyperparam. | Retweet | Amazon | IPTV | SO | Trans. | Mimic | LinkedIn | MemeT. |
|---|---|---|---|---|---|---|---|---|
| Batch size | 20 | 20 | 4 | 20 | 20 | 20 | 20 | 20 |
| Accum Batch | 1 | 1 | 5 | 1 | 1 | 1 | 1 | 1 |
| Embedding size | 32 | 32 | 256 | 32 | 32 | 4 | 256 | 32 |
| Kernel size | 5 | 5 | 5 | 5 | 5 | 7 | 5 | 5 |
| # of filters | 16 | 16 | 16 | 16 | 16 | 16 | 16 | 16 |
| # of layers | 9 | 9 | 2 | 9 | 9 | 2 | 2 | 9 |
| Sim size | 40 | 40 | 46 | 40 | 40 | 46 | 46 | 40 |

Table 6: COTIC training hyperparameters for different datasets. Here Accum Batch is Accumulate grad batches, Embedding size is for event types, the # of filters is the size of inner layers, and Sim size is the number of points for Monte Carlo integration of intensity.

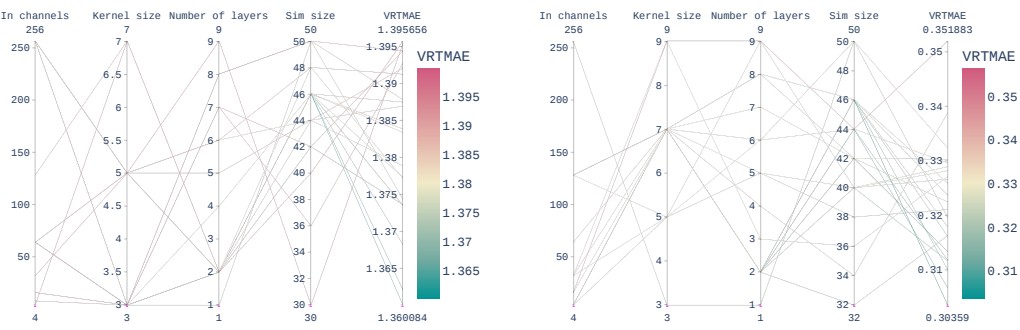

Figure 5: Optuna hyperparameters search for LinkedIn (left) and MIMIC (right) datasets

## C  BASELINE METHODS

**RNN-based models.**  **Neural Hawkes Process** Mei & Eisner (2017a) is one of the first neural models that were proposed for event sequences. This model uses LSTM-like architecture with

additional evolution of the hidden states between events. The main benefit of this model is that it has almost no assumptions on the parametric structure for the intensity interpolation. However, Neural Hawkes is very slow in training. **Recurrent Marked Temporal Point Processes** RMTPP Du et al. (2016) is another RNN-based model proposed for event sequence forecasting. In contrast to Neural Hawkes, this model assumes a parametric structure for the intensity interpolation, which makes it less expressive. Finally, **ODE Temporal Point Process** provides a simplified version from Neural Spatio-Temporal Point Processes Chen et al. (2021), which exploited a different class of parametrization for spatio-temporal point processes. In a nutshell, we model Temporal Point Process with Neural ODE state evolution via an RNN.

**Attention-based models.** **Attentive Neural Hawkes Process** Yang et al. (2022) further develops the idea from the Neural Hawkes Process paper, but instead of an LSTM, they employed Transformer. The generative model, AttNHP, is computationally expensive, as it requires obtaining deep embedding for a possible event type to calculate its intensity at a given timestamp, and during training, likelihood calculation requires computing the intensities of many possible events. **Transformer Hawkes Model** Transformer is a superior architecture for NLP problems. This fact inspired the authors of Zuo et al. (2020) to use transformers in an event sequence set-up. The main benefit of Transformer compared to RNN-like models is that it does not struggle from small memory and vanishing and exploding gradients. However, the solution has some parametric assumptions on the intensity interpolation.

**CNN-based models.** **WaveNet** is a well-known architecture proposed in van den Oord et al. (2016) designed for audio generation. It is one of the prominent time series processing models that adopts the ideas of convolutional neural networks. Efficient implementation and dilated convolutions leading to a large receptive field are among the advantages of WaveNet. Due to these nice properties, we use this convolutional architecture as a baseline. As WaveNet is not directly applicable to event sequence forecasting with non-uniform times between events, we use time lag as an additional input feature. **Continuous CNN for Nonuniform Time Series** In a recent work Shi et al. (2021), it was proposed to use Continuous Convolutional Neural Networks (CCNNs) to model non-uniform time series. Although this method is the closest to the solution we present, it has notable differences and drawbacks regarding event sequence modeling. First, the paper's authors address several problems, i.e. auto-regressive time series forecasting, speech interpolation, and intensity estimation for temporal point processes. However, the architecture itself was designed for signal restoration. In this case, there exists an actual underlying signal that can be defined at any point in time. For this reason, the authors propose to use only one continuous convolutional layer and allocate points evenly after that, proceeding with the standard discrete convolutions. Therefore, the non-uniformity of time exists only in the first layer which does not consider the specifics of the event sequences modeling task. Second, the intensity function is explicitly defined parametrically, strictly limiting the model's expressiveness.

## C.1 Technical details for baseline methods

In general, we used hyperparameters provided by the authors of the methods while adjusting the learning rate for some datasets to improve the performance. Further details are given below.

**RNN-based models.** We use two acknowledged variations of RNN-based models: Neural Hawkes and RMTPP (Recurrent Marked Temporal Point Process). For the *MemeTrack* and *Amazon* datasets, RMTPP time loss quickly explodes within a single epoch, leading to the absence of results, which is reflected in our tables. For **Neural Hawkes**, **RMTPP (Recurrent Marked Temporal Point Process)** we used PyTorch implementation from the following GitHub repo.[1] For a **Neural Spatio-Temporal Point Processes (ODETPP**, which combines RNN and Neural ODE, we have borrowed code from the same repository.

**CNN-based models.** We also adopt two variants of one-dimensional convolutional architectures to process event sequences. For **WaveNet**, we used the following implementation[2]. We note that

---

[1] https://github.com/ant-research/EasyTemporalPointProcess
[2] https://github.com/litanli/wavenet-time-series-forecasting

**WaveNet** does not take into the non-uniformity of a sequence, but it is a strong baseline that takes the roots in CNN ideas. For **CCNN**, following the original architecture[3], we use a model consisting of one continuous convolutional layer followed by standard discrete convolutions and a particular output layer reconstructing intensity in the exponential form given in Shi et al. (2021). Consequently, the likelihood function becomes a double exponent, leading to a robust numerical instability caused by overflows in exponent. To prevent this, we normalize the datasets and use strict early-stopping conditions while training for CCNN.

**Attention-based models.** Finally, we use three SOTA variants of attention-based models specifically tailored to TPP. **THP (Transformer Hawkes Process)** has the implementation from the repo[4]. Subsequently, the model was introduced into the pipeline, in which COTIC training took place. Moreover, we corrected the intensity function and, consequently the likelihood, by excluding the current hidden state to make the model consistent. **THP2SAHP** combines THP Zuo et al. (2020) and the calculation of the intensity function from Zhang et al. (2020) implemented in the repo[5]. Since the intensity (hence, the likelihood) calculation in this approach uses the current hidden state, it cannot be compared with other architectures w.r.t. the likelihood. Finally, we employed realization of **Attentive Neural Hawkes Process** from previously mentioned Ant Research repository.

## D    RANGE OF VALIDITY

A reasonable hypothesis is that the internal characteristics of a dataset affect the selection of the best method for modeling. This experiment is designed to understand how variability within a single dataset affects the quality of the models' performance. We compute the dependence of the target metrics on the maximum sequence length and the number of event types on two datasets — IPTV and LinkedIn. In the first experiment, we limit the length of all sequences by a predefined number and drop the events occurred after that time horizon. For example, if the sequence length is 3100 and the maximal length is 2000, then the last 1100 events are dropped. For the second experiment, we drop less frequent event types from the dataset reducing the total number of present event types.

The results can be found in Figure 6. The COTIC's performance is stable across the considered range of available event types and sequence lengths. Transformers perform worse for long sequences because it is harder to learn a strong attention-based model. The Neural Hawkes model shows unstable results for the return time prediction due to less stable training of LSTM-type architectures.

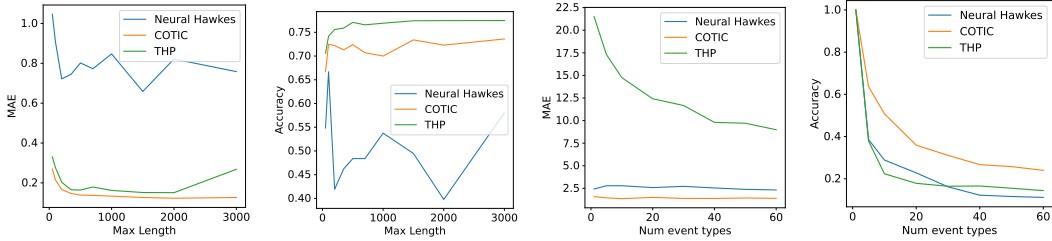

Figure 6: Return time and event type metrics for the different maximal sequence lengths and numbers of event types. The left picture in each pair corresponds to the results for the IPTV dataset; the right picture in each pair corresponds to the results for the LinkedIn dataset.

## E    TRAINING TIME ANALYSIS

Table 7 demonstrates training time per second in similar conditions for considered approaches. There are two clear outsiders — Attentive Neural Hawkes and its predecessor, Neural Hawkes. For this RNN type of model, long training time is explained by heavy parametrization. Our model's result is comparable to the latency of THP algorithms while lagging behind RMTPP (which has simple

---

[3]https://github.com/shihui2010/continuous_cnn
[4]https://github.com/SimiaoZuo/Transformer-Hawkes-Process
[5]https://github.com/QiangAIResearcher/sahp_repo

| Dataset | RMTPP | Neural Hawkes | ODETPP | THP | THP2SAHP | AttNHP | WaveNet | CCNN | COTIC (ours) |
|---------|-------|---------------|--------|-----|----------|--------|---------|------|--------------|
| Retweet | 51.49 | 897.86 | 887.91 | 208.14 | 205.95 | 3868.63 | 1.18 | 86.49 | 185.65 |
| Amazon | – | 1183.38 | 234.13 | 98.71 | 86.91 | 1431.08 | 0.26 | 35.87 | 97.10 |
| IPTV | 30.03 | 170.80 | 178.01 | 159.04 | 165.48 | 11866 | 0.77 | 6.79 | 20.18 |
| SO | 129.085 | 869.97 | 934.59 | 190.25 | 316.32 | 4329.38 | 17.00 | 99.78 | 400.50 |
| Transactions | 1168.32 | 5337.73 | 4396.75 | 310.76 | 316.40 | 38712 | 82.47 | 152.43 | 860.16 |
| Mimic | 0.33 | 0.98 | 2.64 | 1.66 | 1.70 | 0.55 | 0.06 | 0.41 | 1.66 |
| LinkedIn | 1.27 | 1.53 | 4.18 | 10.49 | 10.67 | 0.89 | 2.05 | 2.44 | 11.09 |
| MemeTrack | – | 1610.24 | 518.15 | 1659.73 | 1638.56 | 348.19 | 43.38 | 154.30 | 546.50 |

Table 7: Training time (in seconds) per epoch for each baseline model and our approach COTIC for all eight datasets.

architecture and doesn't work with complex datasets) and WaveNet (being an established benchmark for sequences). Given the fact that the COTIC model provided the best results on downstream tasks and that the closest competitor is Neural Hawkes, in general, our model is shows distinct results if compared to others with respect to quality and efficiency.

