# OpenReview forum: "COTIC: Embracing Non-uniformity in Event Sequence Data via Multilayer Continuous Convolution"
_ICLR.cc/2024/Conference — Submitted to ICLR 2024_

### Official Review · Reviewer_2YVf · 2023-10-19

**Soundness:** 2 fair
**Presentation:** 2 fair
**Contribution:** 2 fair
**Rating:** 3
**Confidence:** 4

**Summary:**

The paper introduces a new model for marked temporal point process data. The proposed model leverages 1-D CNN to capture complex dependence among events and is more efficient than other state of art models on multiple real benchmarks.

**Strengths:**

Originality: somewhat novel, although I have seen a paper with similar idea:
Zhou et al.  Intensity-free convolutional temporal point process: Incorporating local and global event contexts. It is concurrent work.
 The idea is simple to use convolution operator parametrized by neural network as in 1D-CNNs.

Quality: the paper has performed empirical a lot of real benchmarks and demonstrated the claim overall superior performance. The paper also introduced the technical background of TPPs and also the proposed convolution operation in details.

Clarity: the paper is easy to follow from paragraph to paragraph.

Significance: it can be somewhat significant to the TPP community as typical approaches are via neural models for sequences such as RNN and transformer. The 1D CNN approach has some benefits such as computational efficiency compared to RNN and transformer based models.

**Weaknesses:**

Quality: 1. The authors can motivate the use of 1D-CNN more. The authors mentioned transformers have N^2 memory and computational bottleneck. What about 1D-CNN? It would be nice to see any theoretical statement of the complexity (only empirical experiments are demonstrated.)
2. the authors did not include synthetical experiments. It would be nice to include experiments on standard Hawkes, and proximal graphical event models ( Bhattacharjya  et al. proximal graphical event models)  two representative models for modeling real world event sequences in a controlled manner.
3. Empirical experiments and ranks. I am not sure about label prediction. COTIC does not seem to predict well. Could the author provide some insight? Maybe due to the hyperparameter α and β.

Clarity: some parts are not clear to me. “The intensity has to be non-negative and should lay in the space of L1 functions. “ Should it be PDF?  “expected return time ∆ti,k+1 correspondingly via the following formulas:” No formular associated with return time ∆ti,k+1. Did the authors calculated from the PDF?  How do the author make sure  κ(t) = 0 for  t <0  the neural network?

**Questions:**

1.	Some questions are from weakness.

2.	Figure 1: why are two curves in the κ(t)?

3.	Figure 4: Dependence of training time complexity: are the number of parameters comparable for all three models?  Why are only 2 baselines selected? Why are only IPTV and SO used? In this case, some synthetic experiments will be more beneficial where one can control the length of event sequences.

4.	The authors use LogCosh for time prediction. Could other loss functions be used?

5.	What are the hyperparameters for α and β in αLtime + βLtype in the experiments ? If authors are only interested in the prediction, maybe they can choose not to include Lλ in the loss?

---

### Official Review · Reviewer_bk2C · 2023-10-30

**Soundness:** 3 good
**Presentation:** 2 fair
**Contribution:** 2 fair
**Rating:** 3
**Confidence:** 4

**Summary:**

The paper introduces the Continuous-time Convolutional Neural Network (COTIC) for modeling non-uniform event sequences. This method tackles the challenges of methodological non-uniformity and computational intensity in event sequence data by using continuous convolution neural networks with dilations and multi-layer architecture.

**Strengths:**

1. The model's ability to efficiently handle long-term dependencies between events using dilations and a multi-layer architecture addresses a key limitation in many traditional models.

2. COTIC outperforms existing models in predicting event times, indicating its practical strengths in handling real-world event sequence data.

**Weaknesses:**

1. Lack of Novel Technical Contribution: The approach of treating irregular time points with continuous convolutional application echoes methodologies from prior works, like [1]. The proposed method is only around one page.

2. Compared to existing models like Wavenet and the Neural Hawkes model, the improvements, especially in predicting event types, appear to be marginal.

3. The paper's unclear and potentially confusing mathematical notation, like `$L_1$
 functions', can hinder comprehension of the model.

4. The core theoretical innovation of the paper, the convolution theory under non-uniform sequences, is inadequately explained. A more robust formulation of the theoretical background can help better understand and potential application of the model. (Section 3.3)

[1] CKCONV: CONTINUOUS KERNEL CONVOLUTION FOR SEQUENTIAL DATA

**Questions:**

1. What are the potential reasons why COTIC excels at predicting event times but not event types? Does it resemble other irregular time-series-based models that use continuous convolution methods?

2. For the experimental part on time complexity, can the paper provide more details about hyper-parameters, such as batch size, to ensure a fair comparison?

---

### Official Review · Reviewer_etAG · 2023-11-01

**Soundness:** 2 fair
**Presentation:** 1 poor
**Contribution:** 2 fair
**Rating:** 3
**Confidence:** 3

**Summary:**

This paper proposes a new method COTIC for predicting on event sequences with non-uniformity and sparsity. The new CITIC model is empowered by the continous convolution trick. The proposed model is evaluated on several datasets on predicting the next event time and type, which achieves some performance gain over the previous baseline models.

**Strengths:**

1. The targeted problem is important whch involes many applications, and it is a long-standing problem in the research domain (next event time and type prediction on event sequences).
2. The obtained results seems promising with better inference performance on both time prediction and type prediction.
3. The model is also shown to have better empirical running time complexity compared to baseline models.

**Weaknesses:**

1. The paper is hard to read.
    - In Section 3.1, this part seems to be a typo: "for $k > j$ $t_{ik} ≥ t_{jk}$"
    - In Equation (1) and (2): f() is a function on t and m, but S() is defined on t, Could I ask what is the relationship between f() and S()?
    - In Equation (3): It is unclear why the integral of $\lambda_m(t)$ is 1 since it does not seem to be a PDF. Also, could you please inform the reviewer where this information (integral of $\lambda_m(t)$ is 1) is used in the later derivation?
    - How Equation (4) is derived?
    - The reviewer does not follow Section 3.5. It seems that the loss function is independent from the previous content (Sec 3.1 - Sec 3.4).

2. The core contribution -- continuous convolution -- should be emphasized more and explained clearer. What is the kernel $\kappa()$, how is this continuous convolution related to the well-known CNN? Does the proposed model scalable?

3. More experimental details should be specified, such as the MIMIC data has 65 event types, only 285 sequences with average length at 8.6. Then, how is the deep learning model trained? It seems the dataset is too small to train a deep learning model. Similar question for IPTV dataset.

**Questions:**

See the above.

---

### Official Review · Reviewer_jVj6 · 2023-11-01

**Soundness:** 1 poor
**Presentation:** 2 fair
**Contribution:** 2 fair
**Rating:** 3
**Confidence:** 5

**Summary:**

This paper proposed a multi-layer continuous-time convolution framework for modeling asynchronous data from temporal point processes with marks. Instead of using the conditional intensity function, the model predicts future events through MLP based on the learned embedding of history, while the embedding can also be used in other downstream tasks.

**Strengths:**

1. The paper proposes to model the asynchronous event sequences and make predictions based on embedding learned from convolution architectures, a novel perspective other than the commonly-used way of modeling the intensity function.
2. The model shows superior performance in predicting the time and type of the next event, based on results of large-scale datasets.

**Weaknesses:**

First of all, the proposed model has several technical defects in terms of modeling events from temporal point processes:

1. An essential nature of point processes is the stochasticity of future event generation, which means that even with a given history, the occurrences of future events are still stochastic. However, despite the good predictive performance of the proposed model, it only provides deterministic predictions of the next event given history (or to say, the embedding), which leads to a significant model loss when it comes to modeling asynchronous events.
2. The conditional intensity function $\lambda$ of the point process enables such stochastic modeling of future events. However, even though the paper proposes a way to model the $\lambda$, there exists a significant problem. According to equation 7, the evaluation of $\lambda$ at any event time $t_k$ will take the $k$-th event into account. However, according to the nature of point processes [1], the $k$-th event is excluded when evaluating $\lambda(t_k)$. Such a self-inclusion has the severe problem of causality. Also, learning $\lambda$ with self-inclusion based on negative log-likelihood will lead to singular results (extremely large values at event times and small values at non-event times). I expect the authors to have some simulations to prove that their conditional intensity function can accurately recover the ground truth (like the $\lambda$ in the classic Hawkes process with exponentially decaying kernel).

Other weaknesses also exist:

3. No simulation or experiments on synthetic datasets, which makes me suspect whether the model can really recover the ground truth.
4. No results of the comparison of negative log-likelihood on testing sets, which is a classic metric in evaluating the goodness of point process modeling.
4. The notation on page 4 is confusing. If I understand right, $j$ should be the index of events which is an integer, but $T_i$ is the upper limit of the time horizon which is a real number. How $j$ takes the value of $T_i$? And $k > j,  t_{ik} ≥ t_{jk}$ is also confusing.

---
[1] Alan G Hawkes. Spectra of some self-exciting and mutually exciting point processes. Biometrika, 58(1):83–90, 1971.

**Questions:**

1. Could the authors explain why COTIC has a lower time complexity than Neural Hawkes and THP?

---

### Meta-Review · Area_Chair_k2GJ · 2023-12-08

**Metareview:**

This paper's key proposal is modeling asynchronous data using a continuous convolutional neural network (1D) to model the intensity function in a point process. The reviewers on balance felt this was not ready for publication and no responses were made to the reviewer comments. There are several areas for improvement I encourage the authors to consider: First, reconsider the self-inclusion issue in the conditional intensity function raised by reviewer (jVj6) -- this is a severe problem that would void the validity of the generalization results from this model.  Experiment with simulation studies using metrics such as NLL. Improve the clarity of writing and better highlight what the technical innovations herein are.

**Justification For Why Not Higher Score:**

There was consensus to reject this paper among the reviewers (and I agree).

**Justification For Why Not Lower Score:**

N/A

---

### Decision · Program_Chairs · 2024-01-16

Reject